# Sodium Deoxycholate-Propidium Monoazide Droplet Digital PCR for Rapid and Quantitative Detection of Viable *Lacticaseibacillus rhamnosus* HN001 in Compound Probiotic Products

**DOI:** 10.3390/microorganisms12081504

**Published:** 2024-07-23

**Authors:** Ping Wang, Lijiao Liang, Xinkai Peng, Tianming Qu, Xiaomei Zhao, Qinglong Ji, Ying Chen

**Affiliations:** 1Chinese Academy of Inspection and Quarantine, Beijing 100176, China; wangp_129@163.com (P.W.); lianglijiao96@163.com (L.L.); a18140793621@163.com (X.P.); qtm110506@163.com (T.Q.); xiaomei200413@126.com (X.Z.); jql81070305@aliyun.com (Q.J.); 2College of Biosystems Engineering and Food Science, Zhejiang University, Hangzhou 310058, China; 3College of Food Science and Pharmacy, Xinjiang Agricultural University, Urumqi 830052, China; 4College of Food Science and Engineering, Jilin Agricultural University, Changchun 130118, China

**Keywords:** probiotics, number of live bacteria, ddPCR, powder probiotic products, accurate detection

## Abstract

As a famous probiotic, *Lacticaseibacillus rhamnosus* HN001 is widely added to probiotic products. Different *L. rhamnosus* strains have different probiotic effects, and the active HN001 strain is the key to exerting probiotic effects, so it is of great practical significance for realising the detection of *L. rhamnosus* HN001 at the strain level in probiotic products. In this study, strain-specific primer pairs and probes were designed. A combined treatment of sodium deoxycholate (SD) and propidium monoazide (PMA) inhibited the amplification of dead bacterial DNA, establishing a SD-PMA-ddPCR system and conditions for detecting live *L. rhamnosus* HN001 in probiotic powders. Specificity was confirmed using type strains and commercial strains. Sensitivity tests with spiked samples showed a detection limit of 10⁵ CFU/g and a linear quantification range of 1.42 × 10⁵–1.42 × 10⁹ CFU/g. Actual sample testing demonstrated the method’s efficiency in quantifying HN001 in compound probiotic products. This method offers a reliable tool for the rapid and precise quantification of viable *L. rhamnosus* HN001, crucial for the quality monitoring of probiotic products.

## 1. Introduction

Probiotics are live microorganisms that can be beneficial for host health at a certain quantity [1,2]. Accumulating evidence has shown their beneficial health effects on normalising disturbed gut microbiota [3], combating multidrug-resistant pathogens [4], shortening the duration of acute infectious diarrhoea [5], and improving immune function [6]. Accordingly, the global market for probiotics is growing rapidly, and the worldwide market for probiotics is expected to reach USD 77.09 billion by 2025 [7].

*Lacticaseibacillus rhamnosus* HN001 is one of the most widely marketed probiotic strains. Several hundred clinical trials have shown its excellent performance in the modulation of the immune system and in reducing the cumulative prevalence of eczema in early life [8]. Additionally, a combination of *L. rhamnous* HN001 and the phosphoinositide 3-kinase (PI3K)/mTOR dual inhibitor significantly prolongs cardiac transplant survival [9]. These emerging scientific advances have increased the market demand and quality requirements for probiotics. However, many commercial probiotic products are mislabelled or contain an insufficient number of live bacteria [10]. Notably, the health benefits of probiotic products are not only strain-specific, but also dose-dependent [11]. Therefore, the precise identification and quantification of the viable cells of probiotic bacteria at the strain level are crucial, especially in cases where probiotic products are used to help manage inflammatory bowel diseases or disorders of immune function [8].

In previous studies of probiotics, the plate count method and molecular biology were usually used. Plate count methods are easy to use, and their results are intuitive; these methods are widely used in bacteria detection. However, it takes 24 h or more for some probiotics to grow, and these methods are unable to distinguish different strains within a species; thus, they have the disadvantage of a low efficiency in practical applications [12]. Among the molecular biology methods, Zhao et al. developed qPCR-based quantitative assays for *Lacticaseibacillus rhamnosus*, which are able to specifically detect *Lacticaseibacillus rhamnosus* X253 in products, but do not allow for the differentiation of dead bacteria from live bacteria [13]. The use of whole-genome sequencing enables the strain-level detection of probiotic bacteria to elucidate the strain type and function of the probiotic [14]. Although the whole-genome sequencing method can clarify the function of probiotics while achieving strain identification, it is not suitable for frequent testing in industrial production due to its time-consuming and costly nature [15]. In contrast, flow cytometry permits the rapid counting of dead and live cells in a sample by distinguishing between dead and live bacteria, and the screening of target-specific dyes has now become a breakthrough focus in the development of flow cytometry analyses [16].

A rapid FCM counting method for *Lacticaseibacillus* based on three dyes (PI/cFDA, SYTO 24/PI, and DiOC2) was published by the International Organization for Standardization (ISO) and International Dairy Federation (IDF) standards ISO 19344 and IDF 232 [17]. PMA is a fluorescent dye that covalently crosslinks with DNA. Under light conditions, the photosensitive group of PMA is converted into an azabin radical, which covalently cross-links with DNA, thereby blocking the PCR amplification of the DNA molecule. PMA is able to selectively penetrate the cell membranes of dead bacteria and, therefore, it has been used in conjunction with PCR techniques for the quantitative detection of viable bacteria [18]. In a study by Guo et al., live *L. rhamnosus* bacteria were detected in products by combining PMA with qPCR [19]. Droplet digital polymerase chain reaction (ddPCR) is a new method for nucleic acid detection and absolute quantification. Compared with traditional PCR and qPCR, ddPCR splits the DNA sample into a number of droplets, each of which provides a separate digital measurement, making it more accurate and convenient for testing products containing active probiotics. ddPCR quantification does not require a standard curve, is more tolerant for PCR inhibitors, and is widely used in bacteria and virus detection [20]. Combining PMA with ddPCR methods holds great promise for the accurate quantitative detection of live bacteria at the strain level. However, there are few quantitative analyses of live probiotics at the strain level in market samples based on ddPCR. 

In this study, a SD-PMA-ddPCR assay was established for the rapid and quantitative detection of live *L. rhamnosus* HN001 in powder probiotic products. At the same time, the sensitivity and linear quantification range of the SD-PMA-ddPCR method were determined in simulated spiked samples, and the method was also successfully used to detect and quantify *L. rhamnosus* HN001 in commercial products.

## 2. Materials and Methods

### 2.1. Strains and Culture Conditions

*L. rhamnosus* HN001 was obtained from Danisco China Inc; *L. rhamnosus* LGG (ATCC 53103) was purchased from the China General Microbiological Culture Collection Center (CGMCC); *L. rhamnosus* CICC 6135 and *L. rhamnosus* CICC 6151 were purchased from the China Center of Industrial Culture Collection (CICC); and *L. rhamnosus* ATCC 6001, *L. rhamnosus* ATCC 7469, *L. rhamnosus* ATCC 11443, *L. rhamnosus* MP108, *L. casei* ATCC 393, L. casei ATCC 334, L. plantarum CICC 6009, L. plantarum ATCC 8014, *L. delbergini subsp. lactis* CICC 6047, *L. reuteri*, *L. acidophilus* CICC 6074, *L. acidophilus* ATCC 4356, *L. fermentum* ATCC 9338, *L. helveticus* CICC 6032, *B. breve* ATCC 15700, *B. bifidum* ATCC 11863, and *B. bifidum* CICC 6071 were collected from the Chinese Academy of Inspection and Quarantine (IQCC). All *Lactobacillus* and *Bifidobacterium* strains were grown without agitation in Man–Rogosa–Sharpe (MRS) medium (BD Difco, BD Dianostics, Le Pont-De-Claix, France) at 37 °C.

### 2.2. Preparation of Dead Bacteria

To verify the inhibitory effect of the combined treatment of SD and PMA on the DNA amplification of dead bacteria, a heat-inactivated bacterial broth of *L. rhamnosus* HN001 was prepared. According to a previous study [21], the strain was cultured until the cell concentration reached 10^8^ CFU/mL; then, 10 mL of the culture was transferred to a 50 mL centrifuge tube and incubated in an 80 °C water bath under agitation for 10 min, followed by cooling to room temperature. Finally, 1 mL of the heat-inactivated bacterial solution was pipetted, added to three plates with MRS medium, and incubated anaerobically at 37 °C for 48 h for observation. The absence of colony growth on all three plates proved that heat-inactivated cultures were prepared.

### 2.3. Sample Collection and DNA Extraction

Ten powdered probiotic products labelled with *L. rhamnosus* HN001 and other probiotic strain compounds were collected from a market in Beijing (Table 1). Before a probiotic product was extracted for DNA, a 1:10 serial dilution of the sample was prepared. A dilution sample of 1:10 was made by mixing 1 g of probiotic product with 9 mL of 0.1 M phosphate-buffered saline (PBS, pH 7.2) in a BagLight^®^ PolySilk (Interscience, Cantal, France) bag for 2 min to obtain a homogenate. Then, a dilution sample of 1:100 was made by mixing 1 mL of the 1:10 dilution sample with 9 mL of 0.1 M PBS. Finally, a 1:1000 dilution sample was made using a similar method. DNA was extracted from the bacterial cultures and probiotic products by extraction using a Bacterial Genomic DNA Extraction Kit (Tiangen Biotechnology, Beijing, China), according to the manufacturer’s instructions. DNA purity was analysed via UV absorbance using a Nanodrop 2000 spectrophotometer (Thermo Fisher Scientific, Wilmington, MA, USA). The OD260/280 value of the bacterial DNA was between 1.8 and −2.0, and this was considered as suitable.

### 2.4. Primers and Probe Design

The strain-specific gene of *L. rhamnosus* HN001 was obtained using Mauve (version 2.4.0) software according to a method described by Zhang [22]. Based on the specific gene sequences obtained, primers and probes were designed using Primer3 (version 2.6.1) software. The designed primers and probes were blasted on NCBI to verify their theoretical specificity. The final selected pair of specific primers and probe sequences were as follows: the forward primer (FD-F), 5′CATCCAAGCCTTCTCGTGGT3′; the reverse primer (FD-R), 5′ACAACATTTGGTTGGCCTGC3′; and the probe (FD-P); 5′FAM-GCAAGGCCTGCAGAGTAGCGA-BHQ13′.

### 2.5. Screening and Optimisation of ddPCR Assay

ddPCR was conducted on a QX200 Droplet Digital PCR system (BioRad, Hercules, CA, USA), and the reaction mixtures and amplification conditions were screened and optimised. In total, five concentrations of the primers (250, 500, 750, 1000, and 1250 nM), three concentrations of probe (250, 500, and 750 mM), and seven annealing temperatures (56, 57, 58, 59, 60, 61, and 62 °C) were used to determine the best conditions. 

The ddPCR amplification reaction mix contained 10.0 µL of ddPCRTM Supermix for Probes (No dUTP) (Bio-Rad, Hercules, CA, USA), a 1 µL concentration of primer pairs, a 0.5 µL concentration of the probe, and 1 μL of template DNA (20 ng/μL), supplemented to 20.0 µL with DEPC water. The ddPCR amplification cycle conditions were as follows: 10 min of initial denaturation at 95 °C, followed by 40 cycles of incubation of the reaction system at 95 °C for 30 s, different annealing temperatures for 1 min, and 98 °C for 10 min to inactivate the enzyme, with storage at 4 °C. After the ddPCR reaction was completed, a droplet with a fluorescent signal in a droplet analyser was read as a positive droplet, indicating that the droplet contained the target DNA molecule. A droplet with no fluorescence signal was interpreted as a negative droplet, indicating that the droplet did not contain the target DNA molecule or that the content was below the detection limit. The optimal ddPCR parameters were selected based on the clearest boundary between the positive and negative droplets and the largest fluorescence amplitude of the positive droplet.

### 2.6. Optimisation of Detection System for SD-PMA-ddPCR

When quantitatively detecting live bacteria in samples, the DNA of dead bacteria may lead to a false high number in the detection result. Therefore, it is necessary to inhibit the amplification of dead bacterial DNA. A combined treatment with SD and PMA was used to inhibit the effect of dead bacterial DNA on the ddPCR quantitative detection. To obtain the optimal reaction system of SD-PMA-ddPCR for the best amplification inhibition of dead bacterial DNA, different SD concentrations, PMA amounts, and light times were tested, and they were as follows: the PMA amounts were 10.0 mM, 20 mM, 40.0 mM, and 50 mM; the SD final concentrations (*w*/*v*) were 0.04%, 0.08%, 0.1%, and 0.5%; and the light times were 5.0 min, 10.0 min, 15.0 min, and 20.0 min. Orthogonal experiments with 3 factors and 4 levels were conducted according to an L_16_ (34) orthogonal table, with a total of 16 combinations. The experimental results were evaluated according to the detection effect of the SD-PMA-ddPCR method on the dead and live bacterial broths, thus enabling the determination of the optimal additions of SD and PMA and the optimal light time. The optimal additions of SD and PMA and the optimal light time were those that had the best inhibition effects on the amplification of the dead bacterial DNA and little physiological toxicity to the live bacterial DNA.

### 2.7. Specificity of SD-PMA ddPCR

To test the strain level specificity of the SD-PMA-ddPCR assay for *L. rhamnosus* HN001, 7 strains of *L. rhamnosus* and 14 strains of closely related lactic acid bacteria were used. The DNA of these bacteria was extracted using the method in 2.2 as a ddPCR template, with the DNA of *L. rhamnosus* HN001 serving as a positive control. Three parallel experiments were set up for each sample. 

### 2.8. Construction of Standard Plasmid for SD-PMA ddPCR

A recombinant plasmid of the target gene was constructed and used to determine the detection limit and linear quantification range of SD-PMA-ddPCR. According to Liang et al. [23], the FD primer pairs of the target gene were amplified and cloned into the pMD19-T vector (Takara, Osaka, Japan). The plasmid was transformed into *Escherichia coli* JM109 cells and purified using a QIAGEN plasmid mini-extraction kit (Hilden, Germany). The plasmid concentration was determined using a UV spectrophotometer. The extracted plasmid DNA was a tenfold gradient diluted to obtain a 10^8^–10^1^ copies/μL sample to be tested using SD-PMA-ddPCR. 

### 2.9. Practical Detection Limit of the Method

To determine the practical detection limit of the SD-PMA-ddPCR method, simulated spiked milk powder samples were prepared. Briefly, 5 mL of overnight cultured L. rhamnosus HN001 live bacterial solution (10^9^ CFU/mL) and 5 mL of *L. rhamnosus* HN001 dead bacterial solution (10^8^ CFU/mL) were added to 40 mL of a lyophilised protective agent. The lyophilised protectant was formulated as 100 mL of distilled water sterilised with 7 g of skimmed milk powder and 5 g of alginate. Next, 50 mL of the mixed bacterial solution was added to a BagLight^®^ PolySilk bag and fully homogenised for 2 min before being placed in a freezer at −80 °C for 2 h; then, it was lyophilised in a lyophiliser. The working conditions of the lyophiliser were as follows: a vacuum of 17 Pa, a freezing temperature of −40 °C, and a lyophilisation time of 48 h. After freeze-drying, the milk powder spiked samples were fully ground and weighed to simulate commercially available probiotic powder. The number of colony-forming units of *L. rhamnosus* HN001 in the artificially prepared simulated milk powder spiked samples was calculated as shown in Equation (1). The *L. rhamnosus* HN001 solution had a volume of 5 mL, the concentration of the *L. rhamnosus* HN001 solution was 10^9^ CFU/mL, and the powder mass was the mass of the lyophilised powder. A total of 3.5 g of the spiked milk powder sample was obtained after the lyophilisation of 50 mL of the mixed bacterial solution. The number of colony-forming units of *L. rhamnosus* HN001 in the manually prepared simulated milk powder spiked sample was 1.42 × 10^9^ CFU/g.
The number of CFU = (5 mL × 10^9^ CFU/mL)⁄(powder mass (g))(1)

After the artificially spiked samples were prepared, a 1:10 dilution was prepared by taking 1 g of the artificial sample and mixing it with 9 mL of 0.1 M phosphate-buffered saline (PBS, pH 7.2) for 2 min. A 1:100 dilution was then prepared by mixing 1 mL of the 1:10 dilution with 9 mL of 0.1 M PBS. The above steps were repeated to finally obtain 10–10^8^ dilutions of the manually spiked samples. DNA was extracted from 1 mL of the different diluted samples as templates, and the SD-PMA-ddPCR assay was performed to determine the actual detection limit and linear quantification range of the method.

### 2.10. Market Samples Detection

To validate the applicability of the SD-PMA-ddPCR method, ten powdered probiotic products containing *L. rhamnosus* HN001 (labelled with 2–10 different strains per sample) were collected from a market (Table 1). DNA was extracted as described in Section 2.3 and detected using SD-PMA-ddPCR.

## 3. Results

### 3.1. Optimisation of ddPCR Amplification Conditions

The annealing temperature of ddPCR is one of the key parameters affecting the sample performance. To determine the optimal annealing temperature for the SD-PMA-ddPCR detection system, different annealing temperatures (56 °C, 57 °C, 58 °C, 59 °C, 60 °C, 61 °C, and 62 °C) were used and compared (Figure 1A). With an increase in the annealing temperature, the separation of negative and positive droplets gradually increased. When the annealing temperature was 61 °C, the positive and negative droplets were the most clearly demarcated, while the positive droplets had the largest fluorescence amplitude. Therefore, 61 °C was determined as the optimal annealing temperature for the SD-PMA-ddPCR assay. 

The concentrations of primers and probes are also important factors affecting the effectiveness of the ddPCR assay. Five sets of primers and probes with different concentrations were compared (Figure 1B). The fluorescence intensity and the number of microdroplets increased gradually with increases in the primer and probe concentrations. When the primer concentration was 1250 nmol/L and the probe concentration was 500 nmol/L, the highest fluorescence value was observed and there was a better separation area between the negative and positive microdroplet clusters. Therefore, a primer concentration of 1250 nmol/L and a probe concentration of 500 nmol/L were determined to be the optimal concentrations for the SD-PMA-ddPCR assay.

### 3.2. Optimisation of SD, PMA, and Light Time Parameters

Orthogonal experiments were conducted to determine the optimal levels of the relevant parameters of SD, PMA, and light time. The orthogonal table and sample test results are shown in Table 2. In the orthogonal table of three factors and four levels, factor A is the final concentration of PMA, factor B is the amount of SD, and factor C is the light time. The optimised levels were determined from the mean size of an extreme difference analysis by detecting the DNA of the live and dead bacteria as A3, B2, and C3. When the concentration of SD and the amount of PMA in the SD-PMA-ddPCR system were 0.08% (*w*/*v*) and 40 mM and the light time was set to 15 min, the method optimally inhibited the amplification of the dead bacterial DNA while not affecting the normal ddPCR quantitative detection of the live bacterial DNA.

### 3.3. Specificity of SD-PMA-ddPCR Assay

The specificity of the SD-PMA-ddPCR method was verified using 7 strains of *L. rhamnosus*, 11 other strains of Lactobacillus, and 3 strains of *Bifidobacterium* as test subjects. As shown in Figure 2, *L. rhamnosus* HN001 produced fluorescence signals, and no fluorescent signals were seen for the other strains. This indicated that the designed primer pairs, probe, and established SD-PMA-ddPCR method had a good specificity at the strain level and could specifically detect the *L. rhamnosus* HN001 strain.

### 3.4. Sensitivity of the Detection Methods

The sensitivity of the SD-PMA-ddPCR method was determined by constructing recombinant plasmids of the target genes and preparing artificially spiked samples. Recombinant plasmid DNA with copy number concentrations of 1.36–1.36 × 10^8^ copies/μL was used as a template for the SD PMA ddPCR assay (Table 3). Additionally, a linear quantification curve was constructed by comparing the logarithm of the plasmid DNA concentration with the measured values. The results showed that the LOD of the SD-PMA-ddPCR method was as low as 135.81 copies/μL. The quantification range was from 4.2 × 10^−10^ to 4.2 × 10^−6^ ng/μL; the linear quantification equation is shown in Figure 3A. 

The number of colony-forming units of *L. rhamnosus* HN001 in the manually prepared simulated spiked milk powder sample was 1.42 × 10^9^ CFU/g; the calculation method is shown in Equation (1). The simulated spiked samples were subjected to a tenfold gradient dilution to obtain artificially spiked samples of *L. rhamnosus* HN001 at different concentrations. DNA from the different concentrations of the artificially spiked samples was extracted for SD-PMA-ddPCR. Each sample contained dead bacteria as interference in order to confirm the ability of the SD-PMA-ddPCR method to distinguish between dead/live bacteria. The results showed that the actual detection limit of the method was as low as 1.42 × 10^5^ CFU/g (Table 4). The actual quantification of the ddPCR method ranged from 1.42 × 10^5^ to 1.42 × 10^9^ CFU/g (Figure 3B). The SD-PMA-ddPCR assay was able to resist the interference of the dead bacterial DNA in the simulated samples and achieve an accurate quantification of *L. rhamnosus* HN001 live bacteria.

### 3.5. Actual Sample Detection

To determine whether the SD-PMA-ddPCR assay established in this study is suitable for real sample testing, commercially available probiotic powder samples were collected to confirm the method’s suitability. The selected probiotic powder products consisted of infant probiotic powder and adult probiotic powder, with 2–10 different strains labelled per sample. The results showed that the SD-PMA-ddPCR method established in this study successfully detected *L. rhamnosus* HN001 in the multi-strain probiotic products. Meanwhile, the quantitative results of the live bacteria of *L. rhamnosus* HN001 were consistent with the order of magnitude of the product label (Table 1). Sample 10 was a probiotic powder for adults, containing two probiotics (*L. rhamnosus* HN001 and *Bifidobacterium lactis* HN019); the total amount of live probiotics on the label was 10^10^ CFU/1 g, and the detection result was 4.61 ± 0.13 × 10^9^ CFU/g. We speculate that the two probiotics in this product may not have been added in equal amounts. The orders of magnitude of the *L. rhamnosus* HN001 live bacteria detected in most of the probiotic products were consistent with the labels. The probable reason for the detection of Sample 10 showing values lower than those on the label is that the two probiotics were not added in equal amounts in this product. The SD-PMA-ddPCR method established in this study is suitable for the viable detection of *L. rhamnosus* HN001 in powdered probiotic products, and it provides a reliable tool for the rapid and accurate quantitative detection of *L. rhamnosus* HN001 viable bacteria.

## 4. Discussion

Probiotics are widely used in the food and medical fields for their functions in regulating the balance of the intestinal flora, promoting digestion, and enhancing human immunity [2]. Among the various probiotics, *L. rhamnosus* HN001 has unique effects in enhancing human immunity. Recent studies have shown that, in terms of modulating immunity in children, *L. rhamnosus* HN001 significantly reduces the cumulative prevalence of asthma in children at different ages, whereas *Bifidobacterium lactis* HN019 does not [24]. The beneficial effects of probiotics are strain-specific and, in addition to strain characteristics, the efficacy of probiotic products depends on the number of live bacteria [11]. With the expansion of the probiotic industry, governments and industry stakeholders around the world have also stepped up their quality control oversight of commercial probiotic products. Several countries have also set limited standards for the number of live bacteria in probiotic products; for example, Italy, has established a feasible probiotic administration dose of 1 × 10^9^ CFU per day [25], and China requires probiotic health food products with a shelf life to have no fewer than 10^6^ CFU/mL of live bacteria per strain [26]. Currently, a major problem affecting the quality and safety of probiotic products is product mislabelling. The mislabelling of probiotic products may be reflected in the non-compliance of strains or the number of live bacteria [27]. Commercial probiotic products are often supplemented with a variety of active probiotics, and the diversity of probiotics and the stability of the live bacteria in these products pose challenges for quality monitoring. As a result, the mislabelling of commercial probiotic products may be unintentional. In addition to intentional mislabelling, the lack of strain-level probiotic live bacteria detection methods that are rapid, sensitive, and accurate creates difficulties for the regulatory enforcement of relevant authorities and the daily supervision of the probiotic industry. Therefore, there is an urgent need to develop an accurate quantitative method for the strain-level detection of *L. rhamnosus* HN001 in commercial probiotic products.

To solve the problem of probiotic characterisation and mislabelling, related research has gradually shifted from traditional culture methods to high-throughput sequencing. Traditional culture enumeration methods have also been used to achieve the enumeration of target probiotics in commercial products through the development of resuscitative and selective media [28]. High-throughput sequencing assesses the efficacy and potential safety risks of bacteria by sequencing the whole genome of a target strain, enabling the strain-level characterisation of the bacteria along with a gene-based functional annotation [29]. High-throughput sequencing based on metagenomics aims to evaluate the overall composition of the probiotics in complex probiotic products [30]. However, the above research methods have difficulties in balancing between rapid and accurate quantification, strain-level detection, and dead/live bacteria differentiation. As a result, molecular-biology-based assays have attracted researchers’ attention. Among the probiotic assays developed based on molecular biology techniques, the rapid detection of *L. rhamnosus* has been studied: Kim et al. [31] achieved the rapid detection of *L. casei*, *L. paracasei*, and *L. rhamnosus* in a probiotic product via ring-mediated isothermal amplification, with a limit of detection as low as 10^3^ CFU/mL; Wang et al. [32] used an ultrasensitive bacterial blotting electrochemical sensor to determine *L. rhamnosus* GG with a detection limit as low as 5 CFU/mL. However, none of these methods can achieve strain-level *L. rhamnosus* HN001 detection, and it is also difficult to differentiate between dead and live bacteria in test samples. In contrast, in our study, an *L. rhamnosus* HN001-specific single-copy gene fragment was used as the target gene. While achieving strain-level bacterial detection, the ddPCR quantification of the single-copy gene directly corresponded to the precise quantification of *L. rhamnosus* HN001. The detection system was optimised by treating the DNA to be detected with SD coupled with PMA using an orthogonal experimental design. Briefly, the SD-PMA-ddPCR established in this study made it easier for PMA to enter damaged cells through SD pretreatment, and PMA selectively penetrated the cell membrane of the dead bacteria to bind to the DNA and block its PCR amplification. ddPCR detected the unbound DNA of the live bacteria and realised the quantification of the live bacteria; thus, it differentiated between the dead/live bacteria. The theoretical detection limit of the SD-PMA-ddPCR method established in this study was as low as 4.2 × 10^−10^ ng/μL, and the detection limit of the simulated samples was as low as 1.42 × 10^5^ CFU/g. For a practical applicability study of the samples, ten powder probiotic products were selected for testing. The number of probiotic species in the complex probiotic powders ranged from two to ten, with interference from other strains of *L. rhamnosus*. The results of the applicability study showed that the established SD-PMA-ddPCR method was able to rapidly and accurately detect the number of viable bacteria of *L. rhamnosus* HN001 in commercial probiotic powders, and the whole process of the assay took less than 1 h. To the best of our knowledge, the method established in the present study achieved the rapid quantitative detection of the viable bacteria of *L. rhamnosus* HN001 at the strain level for the first time. This method is a reliable tool for market supervision by relevant departments and daily monitoring in the probiotic industry, and it can effectively identify mislabelling in probiotic products labelled with *L. rhamnosus* HN001, as well as safeguard their quality and safety.

## 5. Conclusions

In this study, an accurate quantitative SD-PMA-ddPCR method for the detection of viable *L. rhamnosus* HN001 bacteria in powdered probiotic products was successfully established. The method can achieve the accurate quantification of target bacteria without pre-enrichment treatment. The practical detection limit of the established method was as low as 1.42 × 10^5^ CFU/g, and the linear quantification range was between 1.42 × 10^5^ and 1.42 × 10^9^ CFU/g. To the best of our knowledge, this is the first rapid and accurate quantification of *L. rhamnosus* HN001 at the strain level with the ability to specifically identify the live bacteria in a probiotic product. The established method is of great significance for the quality monitoring of probiotic products containing *L. rhamnosus* HN001. In addition, the SD-PMA ddPCR method has a broader applicability and can be extended for the detection of other probiotic strains. This provides a versatile and flexible quality control tool for the probiotic industry.

## Figures and Tables

**Figure 1 microorganisms-12-01504-f001:**
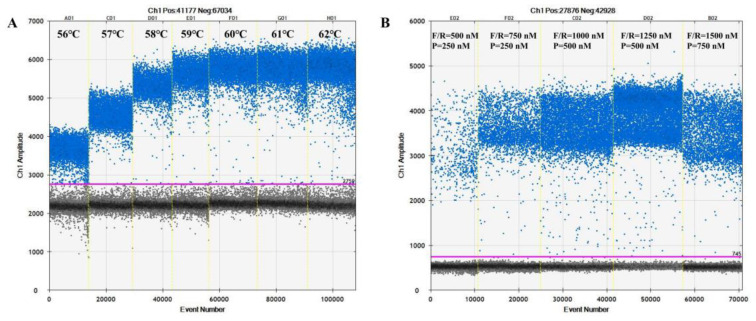
(**A**). Optimised results for ddPCR annealing temperature (**B**). Optimised results of ddPCR primer and probe concentrations. The blue dots represent positive drops, and the grey dots represent negative drops. The purple line marks the boundary between positive droplets and negative droplets.

**Figure 2 microorganisms-12-01504-f002:**
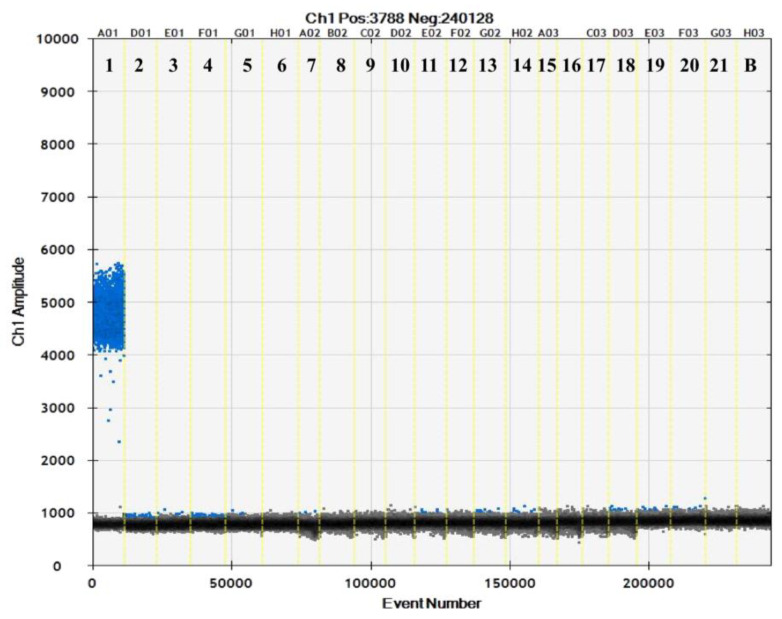
Validation results of primers and probes for *Lacticaseibacillus rhamnosus* HN001. The blue dots represent positive drops, and the grey dots represent negative drops. 1. *L. rhamnosus* HN001; 2–8. Other *L. rhamnosus*; 9–18. Other *Lactobacillus spp.*; 19–21. *Bifidobacterium spp*.; B. Blank.

**Figure 3 microorganisms-12-01504-f003:**
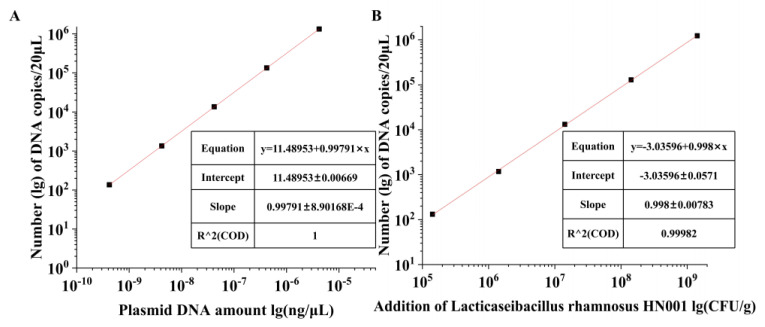
Linear regression of the SD-PMA-ddPCR assay. (**A**). Copy numbers vs. plasmid DNA amount(ng/μL). (**B**) Copy numbers vs. Addition amount of *L. rhamnosus* HN001(CFU/mL).

**Table 1 microorganisms-12-01504-t001:** Characterisation of samples used in this study and the number of live *Lacticaseibacillus rhamnosus* HN001 bacteria detected in commercially available probiotic complex samples.

Sample No.	Retailers	Description	Number of Probiotic Strains Added	Total Amount of Probiotics on the Label of Probiotic Products	Detection Value 1 of *L. rhamnosus* HN001 Live Bacteria (CFU/g)	Detection Value 2 of *L. rhamnosus* HN001 Live Bacteria (CFU/g)	Detection Value 3 of *L. rhamnosus* HN001 Live Bacteria (CFU/g)	Average Values (CFU/g)
1	I	Children’s Probiotic Powder	2	10^10^ CFU/1.5 g	2.79 × 10^9^	2.44 × 10^9^	2.57 × 10^9^	2.6 ± 0.18 × 10^9^
2	II		5	10^10^ CFU/1.5 g	1.24 × 10^9^	1.12 × 10^9^	1.18 × 10^9^	1.18 ± 0.06 × 10^9^
3	III		4	6.0 × 10^9^ CFU/2 g	6.96 × 10^8^	6.63 × 10^8^	7.14 × 10^8^	6.91 ± 0.26 × 10^8^
4	IV		4	7.5 × 10^9^ CFU/2 g	7.29 × 10^8^	6.92 × 10^8^	6.66 × 10^8^	6.96 ± 0.31 × 10^8^
5	V		10	3 × 10^10^ CFU/2.25 g	1.03 × 10^9^	8.2 × 10^8^	1.15 × 10^9^	1.04 ± 0.17 × 10^9^
6	I	Probiotic Powder for Adults	4	10^10^ CFU/2 g	1.09 × 10^9^	9.12 × 10^8^	9.88 × 10^8^	0.99 ± 0.09 × 10^9^
7	VI		6	10^10^ CFU/0.5 g	1.44 × 10^9^	1.42 × 10^9^	1.56 × 10^9^	1.47 ± 0.08 × 10^9^
8	II		2	4.5 × 10^7^ CFU/1.5 g	1.21 × 10^7^	1.30 × 10^7^	1.13 × 10^7^	1.21 ± 0.09 × 10^7^
9	VII		10	2 × 10^10^ CFU/1 g	1.75 × 10^9^	1.61 × 10^9^	1.87 × 10^9^	1.74 ± 0.13 × 10^9^
10	IV		2	10^10^ CFU/1 g	4.66 × 10^9^	4.46 × 10^9^	4.72 × 10^9^	4.61 ± 0.13 × 10^9^

Number of live *L. rhamnosus* HN001 bacteria in probiotic powder (CFU/g) = Measured value (copies/20 μL) × Total volume of extracted DNA (100 μL) × Dilution of probiotic powder.

**Table 2 microorganisms-12-01504-t002:** Three-factor and four-level orthogonal table for optimizing the SD-PMA-ddPCR assay system.

Test No.	Factors			Protocol	Measured Values (Copies/μL)
	**A**	**B**	**C**		
1	50	0.08	10.00	A4B2C2	Live: 9.33 × 10^7^ ± 4.73 × 10^5^ Dead: 5.54 × 10^3^ ± 1.56 × 10^2^
2	40	0.10	5.00	A3B3C1	Live: 9.57 × 10^7^ ± 3.51 × 10^5^ Dead: 1.44 × 10^5^ ± 6.11 × 10^3^
3	10	0.04	5.00	A1B1C1	Live: 9.64 × 10^7^ ± 3.00 × 10^5^ Dead: 8.75 × 10^7^ ± 1.17 × 10^6^
4	20	0.10	15.00	A2B3C3	Live: 9.33 × 10^7^ ± 6.24 × 10^5^ Dead: 1.08 × 10^6^ ± 3.61 × 10^4^
5	50	0.50	5.00	A4B4C1	Live: 9.00 × 10^7^ ± 7.02 × 10^5^ Dead: 8.61 × 10^5^ ± 2.27 × 10^4^
6	40	0.50	20.00	A3B4C4	Live: 9.14 × 10^7^ ± 1.47 × 10^6^ Dead: 3.14 × 10^3^ ± 9.29 × 10^1^
7	40	0.08	15.00	A3B2C3	Live: 9.71 × 10^7^ ± 4.36 × 10^5^ Dead: 2.57 × 10^2^ ± 1.45 × 10^1^
8	20	0.08	5.00	A2B2C1	Live: 9.63 × 10^7^ ± 7.81 × 10^5^ Dead: 4.31 × 10^7^ ± 1.99 × 10^6^
9	20	0.50	10.00	A2B4C2	Live: 9.12 × 10^7^ ± 1.23 × 10^6^ Dead: 1.53 × 10^7^ ± 1.46 × 10^6^
10	40	0.04	10.00	A3B1C2	Live: 9.69 × 10^7^ ± 4.36 × 10^5^ Dead: 4.38 × 10^3^ ± 2.51 × 10^2^
11	50	0.10	20.00	A4B3C4	Live: 9.12 × 10^7^ ± 1.58 × 10^6^ Dead: 2.08 × 10^4^ ± 1.78 × 10^3^
12	50	0.04	15.00	A4B1C3	Live: 9.25 × 10^7^ ± 5.57 × 10^5^ Dead: 5.98 × 10^5^ ± 1.01 × 10^6^
13	20	0.04	20.00	A2B1C4	Live: 9.46 × 10^7^ ± 1.38 × 10^6^ Dead: 1.73 × 10^7^ ± 7.09 × 10^5^
14	10	0.10	10.00	A1B3C2	Live: 9.53 × 10^7^ ± 5.69 × 10^5^ Dead: 8.15 × 10^7^ ± 7.77 × 10^5^
15	10	0.50	15.00	A1B4C3	Live: 9.32 × 10^7^ ± 4.16 × 10^5^ Dead: 7.90 × 10^7^ ± 4.00 × 10^5^
16	10	0.80	20.00	A1B2C4	Live: 9.38 × 10^7^ ± 3.51 × 10^5^ Dead: 7.42 × 10^7^ ± 1.17 × 10^6^

**Table 3 microorganisms-12-01504-t003:** Theoretical values, measured values, and relative standard deviations calculated by quantitative equations after quantification of *Lacticaseibacillus rhamnosus* HN001.

DNA Sample (ng/μL)	Theoretical Values (Copies/20 μL)	Measured Values 1 (Copies/20 μL)	Measured Values 2 (Copies/20 μL)	Measured Values 3 (Copies/20 μL)	Average Values (Copies/20 μL)	SD (Copies/20 μL)	RSD (%)
4.2 × 10^−6^	1,358,962.61	1,328,645.20	1,318,002.25	1,327,056.61	1,324,568.02	4687.78	0.35%
4.2 × 10^−7^	135,896.26	133,642.63	134,413.82	134,121.00	134,059.15	317.86	0.24%
4.2 × 10^−8^	13,589.62	13,863.62	13,227.80	13,524.83	13,538.75	259.76	1.92%
4.2 × 10^−9^	1358.96	1404.82	1282.10	1337.80	1341.57	50.17	3.74%
4.2 × 10^−10^	135.81	140.63	131.84	134.42	135.63	3.69	2.72%

**Table 4 microorganisms-12-01504-t004:** Quantification of Lacticaseibacillus rhamnosus HN001 in artificially spiked samples.

Concentration of *L. rhamnosus* HN001 in Manually Spiked Samples (CFU/g)	Measured Value 1 (Copies/20 μL)	Measured Value 2 (Copies/20 μL)	Measured Value 3 (Copies/20 μL)	Average Values (Copies/20 μL)	Measured Value Conversion Results (CFU/g)	RSD (%)
1.42 × 10^4^	-	-	-	-	-	-
1.42 × 10^5^	128.04	131.83	135.62	131.83 ± 3.09	1.33 × 10^5^	2.35
1.42 × 10^6^	1154.88	1211.65	1158.18	1174.90 ± 26.02	1.17 × 10^6^	2.21
1.42 × 10^7^	13,082.33	13,755.46	12,706.92	13,182.57 ± 433.78	1.32 × 10^7^	3.29
1.42 × 10^8^	125,638.19	129,616.37	131,220.32	128,824.96 ± 2346.60	1.28 × 10^8^	1.82
1.42 × 10^9^	1,175,519.67	1,287,043.22	1,228,243.42	1,230,268.77 ± 45,551.82	1.23 × 10^9^	3.70

The ddPCR results were measured values (copies/20 μL). In this study, 100 μL of DNA was extracted using 1 mL of artificially prepared bacterial solution. Therefore, measured value conversion results (CFU/mL) = Measured values (copies/20 μL) × 100 μL × Sample dilution degree.

## Data Availability

The original contributions presented in the study are included in the article, further inquiries can be directed to the corresponding author.

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
