# Peer review of "Sodium Deoxycholate-Propidium Monoazide Droplet Digital PCR for Rapid and Quantitative Detection of Viable Lacticaseibacillus rhamnosus HN001 in Compound Probiotic Products"

_microorganisms, 2024, doi:10.3390/microorganisms12081504_

Round 1

Reviewer 1 Report

Comments and Suggestions for Authors

General Comments:

The work deserves recognition and relevance, and it has the potential to be well-received by the journal Microorganisms. However, some details need to be revised. I agree that the manuscript should be accepted after some adjustments, as outlined below.

Keywords: Avoid using words that are already in the title.

In the conclusion, the authors need to focus on closing the article. Avoid starting the text with "in summary." A conclusion should be simple, direct, and objective.

Author Response

Thank you for your suggestion. We have modified some of the vocabulary in the key words by deleting "propidium monoazide" and modifying digital PCR to "ddPCR". L. rhamnosus HN001 was retained in the key words without modification because it was the target bacterium in this study.

Reviewer 2 Report

Comments and Suggestions for Authors

The manuscript „A Sodium Deoxycholate-Propidium Monoazide Droplet Digital PCR for Rapid and Quantitative Detection of Viable Lacticaseibacillus Rhamnosus HN001 in Compound Probiotic Products“ by Wang and coauthors describes SD-PMA-ddPCR method for quantification of Lacticacaseibacillus rhamnosus that distinguishes live from dead bacteria. The authors claim in the first sentence of the abstract that, so far there are no DNA-based quantification methods for Lacticacaseibacillus rhamnosus. However, there are already qPCR-based methods for L. rhamnosus (10.3390/foods11152282, https://doi.org/10.7883/yoken.JJID.2019.102) and also another PCR method that also uses PMA and is specific to the HN001 strain as well (10.3389/fmicb.2024.1341884). Therefore, the method described here could only be defined as the first ddPCR method for the HN001 strain. However, the rationale behind developing such a method should then be explained, considering the above-mentioned existing methods. The authors should cite all of these papers and be more precise in clarifying what distinguishes their method among others.

The quality of Figure 3 should be improved so that tables can be read without magnification.

Comments on the Quality of English Language

The authors should also check English again. For instance, the sentence in lines 46-49 should read “Therefore, the precise identity and quantification of the viable cells of the probiotic bacteria at the strain level is crucial…” instead “quantitative”.

In line 63 “achieving” is better word choice than “realizing”. There were also few instances below in the text where “achieving” was better word choice than “realizing”.

In line 76 “for rapid and quantification detection” should be corrected to “rapid and quantitative” detection.

Author Response

  1. Q: The authors claim in the first sentence of the abstract that, so far there are no DNA-based quantification methods for Lacticacaseibacillus rhamnosus. However, there are already qPCR-based methods for L. rhamnosus (10.3390/foods11152282, https://doi.org/10.7883/yoken.JJID.2019.102) and also another PCR method that also uses PMA and is specific to the HN001 strain as well (10.3389/fmicb.2024.1341884). Therefore, the method described here could only be defined as the first ddPCR method for the HN001 strain. However, the rationale behind developing such a method should then be explained, considering the above-mentioned existing methods. The authors should cite all of these papers and be more precise in clarifying what distinguishes their method among others.

AU: Thank you for your suggestion. We have modified the abstract section to add the reason why we want to develop a test for Lactobacillus rhamnosus HN001 viable bacteria. Also, the citation of literature (10.3390/foods11152282; 10.3389/fmicb.2024.1341884) in the introduction section clarifies the difference between the method established in this study and other methods.

  1. Q: The quality of Figure 3 should be improved so that tables can be read without magnification.

AU: Yes, we have revised it.

  1. Q: The authors should also check English again. For instance, the sentence in lines 46-49 should read “Therefore, the precise identity and quantification of the viable cells of the probiotic bacteria at the strain level is crucial…” instead “quantitative”.

AU: Yes, we have revised it.

  1. Q: In line 63 “achieving” is better word choice than “realizing”. There were also few instances below in the text where “achieving” was better word choice than “realizing”.

AU: Yes, we have revised it.

  1. Q: In line 76 “for rapid and quantification detection” should be corrected to “rapid and quantitative” detection.

AU: Yes, we have revised it.

  1. Q: In the conclusion, the authors need to focus on closing the article. Avoid starting the text with "in summary." A conclusion should be simple, direct, and objective.

AU: Thank you for your suggestion. We emphasise the importance of the method established in this study in product quality control. We have also illustrated that this study is expected to increase the number of bacteria detected and further enhance the applicability of the method, thus assisting the development of the probiotic industry.

Reviewer 3 Report

Comments and Suggestions for Authors

The paper of Ping Wang described a reliable tool for rapid and accurate quantitative detection of L. rhamnosus HN001 viable bacteria, for quality monitoring of powder probiotic products containing L. rhamnosus HN001. I believe that  this paper is very interesting and useful in order to draw up an experimental protocol to be followed for the development of the determination of even traces of microorganisms. The development of the ddPCR for the detection of  L. rhamnosus HN001 I think it can be a suitable choice, as it is a technique this much more sensitive than qPCR.

Author Response

Thank you for your approval.

Reviewer 4 Report

Comments and Suggestions for Authors

The manuscript addresses the urgent need for a reliable and strain-specific method to quantify live Lacticaseibacillus rhamnosus HN001 in probiotic products. The developed SD-PMA ddPCR assay has promising characteristics including high sensitivity, specificity and applicability to real samples. The manuscript is well structured, clearly written and supported by sufficient data. Investigating the applicability of the SD-PMA-ddPCR approach to quantify other probiotic strains or bacterial species would expand the potential impact of the method. Additional validation studies with a larger and more diverse set of probiotic products would strengthen the generalizability of the results.

·         Introduction – There is a lack of a clear connection between the limitations of existing methods and the introduction of SD-PMA-ddPCR. Mention how this method addresses the shortcomings of traditional methods. The sentence about combining PMA with qPCR seems inappropriate and can be deleted. Explain the concept of propidium monoazide (PMA) viability staining and its role in distinguishing live from dead cells within the SD-PMA ddPCR method. The text does not explicitly mention why targeting viable cells is crucial. Consider adding a sentence or two about the advantages of ddPCR over qPCR for absolute quantification, especially in the context of analyzing complex samples such as probiotic products.

·         Materials and Methods – Some details about the SD-PMA treatment itself (concentration, volumes used) are missing. Describe the SD-PMA treatment procedure, including the concentrations and volumes of SD and PMA used and the light source/wavelength for PMA photoactivation. There is no explicit mention of how the ddPCR system distinguishes positive and negative droplets (fluorescence). In Section 2.5, mention that ddPCR distinguishes positive and negative droplets based on fluorescence intensity after amplification. Equation (1) could use a brief explanation. Explain Equation (1) to clarify how CFU/g in the spiked powder is calculated.

·         Results – A brief explanation of how ddPCR distinguishes positive and negative droplets (fluorescence intensity) is missing. In Section 3.1, mention that ddPCR distinguishes positive and negative droplets based on fluorescence intensity after amplification. The discussion of Equation (1) in Section 3.4 can be moved to the Materials and Methods section, where the equation is introduced. Consider adding a short concluding sentence at the end of Section 3.5 to summarize the key results of the real sample analysis.

·         Discussion – Consider briefly mentioning how the SD-PMA ddPCR method distinguishes dead/live bacteria. For clarity, consider replacing “high-throughput histology” with “high-throughput sequencing” in the first paragraph.

·         Conclusions – Emphasize the potential of the developed method to improve quality control practices in the probiotic industry and ensure consumer protection. Consider adding a sentence about the potentially broader applicability of the SD-PMA ddPCR method beyond the specific strain (L. rhamnosus HN001). Consider adding a future direction or possible application of the method. For example, you could mention how this method could be adapted to detect other probiotic strains or integrated into routine quality control workflows.

·         Abstract – Lacks conciseness – can be shortened while retaining important information. The sentence about checking specificity can be reworded for clarity. The term “accurate detection” is redundant (already implied by quantification). Define SD (sodium deoxycholate) at the first mention.

Comments on the Quality of English Language

 Moderate editing of English language required.

Author Response

  1. Q: Introduction – There is a lack of a clear connection between the limitations of existing methods and the introduction of SD-PMA-ddPCR. Mention how this method addresses the shortcomings of traditional methods. The sentence about combining PMA with qPCR seems inappropriate and can be deleted. Explain the concept of propidium monoazide (PMA) viability staining and its role in distinguishing live from dead cells within the SD-PMA ddPCR method. The text does not explicitly mention why targeting viable cells is crucial. Consider adding a sentence or two about the advantages of ddPCR over qPCR for absolute quantification, especially in the context of analyzing complex samples such as probiotic products.

AU: Thank you for your suggestion. We have added the rationale for PMA inhibition of DNA amplification in dead bacteria to the introduction section, as well as supplementing the existing information on the current state of Lactobacillus rhamnosus detection methods to make clear the advantages of ddPCR over qPCR and to highlight the significance of the method established in this study.

  1. Q: Materials and Methods – Some details about the SD-PMA treatment itself (concentration, volumes used) are missing. Describe the SD-PMA treatment procedure, including the concentrations and volumes of SD and PMA used and the light source/wavelength for PMA photoactivation. There is no explicit mention of how the ddPCR system distinguishes positive and negative droplets (fluorescence). In Section 2.5, mention that ddPCR distinguishes positive and negative droplets based on fluorescence intensity after amplification. Equation (1) could use a brief explanation. Explain Equation (1) to clarify how CFU/g in the spiked powder is calculated.

AU: Thank you for your suggestion. The parameters used for SD-PMA processing are noted in Section 3.6 and Table Since the volume of the different detection systems varies with the instrument, the final concentration nM (nmol/l) is used as a unit in the text instead of the concentration or volume.

At the end of PCR amplification, the ddPCR system transfers the PCR reaction plate to a microdroplet analyser, where fluorescent signals are detected for each microdroplet. These fluorescent signals come from the specific binding of the target DNA molecule in the microdroplet to the fluorescent probe. If the microdroplet contains a target DNA molecule that has been amplified by PCR, the fluorescent probe will be cut off, thus emitting a fluorescent signal. A microdroplet with a fluorescent signal is read as a positive droplet, indicating that the microdroplet contains the target DNA molecule. Droplets with no fluorescent signal are interpreted as negative droplets, indicating that the droplet does not contain the target DNA molecule or the content is below the detection limit. In addition, we have added this section in 2.5. A short explanatory note has been added to Equation (1). The discussion of equation (1) in Section 3.4 has been transferred to Section 2.9.

  1. Q: Results – A brief explanation of how ddPCR distinguishes positive and negative droplets (fluorescence intensity) is missing. In Section 3.1, mention that ddPCR distinguishes positive and negative droplets based on fluorescence intensity after amplification. The discussion of Equation (1) in Section 3.4 can be moved to the Materials and Methods section, where the equation is introduced. Consider adding a short concluding sentence at the end of Section 3.5 to summarize the key results of the real sample analysis.

AU: Thank you for your suggestion. Instructions on how to distinguish between positive and negative droplets (fluorescence intensity) have been added to Section 2.5. The discussion of equation (1) in Section 3.4 has been transferred to Section 2.9 of Materials and Methods. A short concluding sentence (The orders of magnitude of Lactobacillus rhamnosus HN001 live bacteria detected in most of the probiotic products were consistent with the labels. The probable reason that Sample 10 was detected at less than the labelled additions is that the two probiotics were not added in equal amounts in this product) has been added at the end of Section 3.5.

  1. Q: Discussion – Consider briefly mentioning how the SD-PMA ddPCR method distinguishes dead/live bacteria. For clarity, consider replacing “high-throughput histology” with “high-throughput sequencing” in the first paragraph.

AU: Thank you for your suggestion. We have briefly described the principle of how the SD-PMA ddPCR method differentiates between dead/live bacteria in the discussion. We have replaced "high-throughput histology" with "high-throughput sequencing".

  1. Q: Conclusions – Emphasize the potential of the developed method to improve quality control practices in the probiotic industry and ensure consumer protection. Consider adding a sentence about the potentially broader applicability of the SD-PMA ddPCR method beyond the specific strain (L. rhamnosus HN001). Consider adding a future direction or possible application of the method. For example, you could mention how this method could be adapted to detect other probiotic strains or integrated into routine quality control workflows.

AU: Thank you for your suggestion. We emphasise the importance of the method established in this study in product quality control. We have also illustrated that this study is expected to increase the number of bacteria detected and further enhance the applicability of the method, thus assisting the development of the probiotic industry.

  1. Q: Abstract – Lacks conciseness – can be shortened while retaining important information. The sentence about checking specificity can be reworded for clarity. The term “accurate detection” is redundant (already implied by quantification). Define SD (sodium deoxycholate) at the first mention.

AU: We have shortened the abstract by removing repetitive narratives of method specificity.

Round 2

Reviewer 2 Report

Comments and Suggestions for Authors

The authors have made a lot of improvements to the manuscript in all of its parts. However, the novelty of their research is questionable, as there are already methods in literature based on qPCR and the usage of propidium monoazide for the detection of the same L. rhamnosus strain.

Comments on the Quality of English Language

Only minor English changes are necessary. The whole text should be carefully checked again.

Author Response

Thank you for providing the references. We carefully read and compared the study of Guo et.al., (2024). The robust PMA-qPCR tool developed by Guo et al. enabled the detection of live bacteria of L. rhamnosus for all L. rhamnosus including L. rhamnosus HN001 (non-strain level). The inclusivity and exclusivity of the primers demonstrated its high specificity to L. rhamnosus, which allows accurate identification of the target bacteria. Provide possible application of the PMA-qPCR method to industry for viable cell numeration of L. rhamnosus in compound probiotic products.The health functions of probiotic products are strain-specific, i.e., probiotics of different subspecies of the same species have different probiotic functions (Chen et.al., 2024). Therefore, it is of great significance to development of strain-level identification of L. rhamnosus HN001. In addition, our study was based on the SD-PMA-ddPCR method developed for the single-copy gene of L. rhamnosus HN001, can achieve accurate quantification of L. rhamnosus HN001 viable bacteria at the strain level without the need for a standard curve, which provides a reliable tool for market regulation by the relevant authorities and daily monitoring of the probiotic industry.

Guo, L., Ze, X., Feng, H., Liu, Y., Ge, Y., Zhao, X., ... & Yao, S. (2024). Identification and quantification of viable Lacticaseibacillus rhamnosus in probiotics using validated PMA-qPCR method. Frontiers in Microbiology, 15, 1341884.

Chen, J., Zhang, J., Xie, M., Hao, Q., Liang, H., Li, M., ... & Zhou, Z. (2024). The effect of dietary supplementation with Lactobacillus rhamnosus GCC-3 fermentation product on gut and liver health of common carp (Cyprinus carpio). Aquaculture Reports, 35, 101983.

Reviewer 4 Report

Comments and Suggestions for Authors

The manuscript has been effectively revised and enhanced by the authors. All my suggestions were well taken into account by the authors. The authors have adeptly addressed previous comments, leading to an overall enhancement of the manuscript. I do not have any questions. Therefore, this manuscript may be considered for publication in this journal.

Author Response

Thank you for your approval.